# Dynamics of G6PD activity in patients receiving weekly primaquine for therapy of *Plasmodium vivax* malaria

Walter R. J. Taylor [1,2,3] *, Saorin Kim[4], Sim Kheng[1], Sinoun Muth[1], Pety Tor[4], Eva Christophel[5], Mavuto Mukaka[3,6], Alexandra Kerleguer[4], Lucio Luzzatto[7,8], J. Kevin Baird[6,9●], Didier Menard [10,11,12,13●]

1 National Center for Parasitology, Entomology and Malaria Control, Phnom Penh, Cambodia, 2 Service de Médecine Tropicale et Humanitaire, Hôpitaux Universitaires de Genève, Genève, La Suisse, 3 Mahidol Oxford Tropical Medicine Research Unit, Bangkok, Thailand, 4 Institut Pasteur du Cambodge, Phnom Penh, Cambodia, 5 WHO Western Pacific Regional Office, Manila, Philippines, 6 Centre for Tropical Medicine & Global Health, University of Oxford, Oxford, United Kingdom, 7 Muhimbili University of Health and Allied Sciences, Dar-es-Salaam, Tanzania, 8 Università di Firenze, Florence, Italy, 9 Eijkman Oxford Clinical Research Unit, Eijkman Institute of Molecular Biology, Jakarta, Indonesia, 10 Malaria Genetics and Resistance unit, Institut Pasteur, Paris, France, 11 INSERM U1201, Paris, France, 12 Institut de Parasitologie et Pathologie Tropicale, UR7292 Dynamique des interactions hôte pathogène, Fédération de Médecine Translationnelle, Université de Strasbourg, Strasbourg, France, 13 Laboratoire de Parasitologie et Mycologie Médicale, Les Hôpitaux Universitaires de Strasbourg, Strasbourg, France

● These authors contributed equally to this work.
* bob@tropmedres.ac

**Data Availability Statement:** Selected data generated and analysed during this study are included in this published article and its Supplementary information files. Requests for

## Abstract

### Background

Acute *Plasmodium vivax* malaria is associated with haemolysis, bone marrow suppression, reticulocytopenia, and post-treatment reticulocytosis leading to haemoglobin recovery. Little is known how malaria affects glucose-6-phosphate dehydrogenase (G6PD) activity and whether changes in activity when patients present may lead qualitative tests, like the fluorescent spot test (FST), to misdiagnose G6PD deficient (G6PDd) patients as G6PD normal (G6PDn). Giving primaquine or tafenoquine to such patients could result in severe haemolysis.

### Methods

We investigated the G6PD genotype, G6PD enzyme activity over time and the baseline FST phenotype in Cambodians with acute *P. vivax* malaria treated with 3-day dihydroartemisinin piperaquine and weekly primaquine, 0·75 mg/kg x8 doses.

### Results

Of 75 recruited patients (males 63), aged 5–63 years (median 24), 15 were G6PDd males (14 Viangchan, 1 Canton), 3 were G6PD Viangchan heterozygous females, and 57 were G6PDn; 6 patients had α/β-thalassaemia and 26 had HbE.

Median (range) Day0 G6PD activities were 0·85 U/g Hb (0·10–1·36) and 11·4 U/g Hb (6·67–16·78) in G6PDd and G6PDn patients, respectively, rising significantly to 1·45 (0·36–5·54, p<0·01) and 12·0 (8·1–17·4, p = 0·04) U/g Hb on Day7, then falling to ~Day0 values by

additional data can be made in the first instance to the data access committee (datasharing@tropmedres.ac) that considers requests for data.

**Funding:** EC organised the funding of the study from WHO Headquarters. There is no reference number. At the time of the study, WRJT was part supported by France Expertise International through the 5% initiative as a consultant to CNM in operational research. JKB was supported by Wellcome Trust grant B9RJIXO and DM by the French Ministry of Foreign Affairs. S Kim was supported by an APMEN fellowship grant (103-09). The funder had no role in study design, data collection and analysis, decision to publish, or preparation of the manuscript.

**Competing interests:** The authors have declared that no competing interests exist.

Day56. Day0 G6PD activity did not correlate (p = 0.28) with the Day0 reticulocyte counts but both correlated over time. The FST diagnosed correctly 17/18 G6PDd patients, misclassifying one heterozygous female as G6PDn.

## Conclusions

In Cambodia, acute *P. vivax* malaria did not elevate G6PD activities in our small sample of G6PDd patients to levels that would result in a false normal qualitative test. Low G6PDd enzyme activity at disease presentation increases upon parasite clearance, parallel to reticulocytosis. More work is needed in G6PDd heterozygous females to ascertain the effect of *P. vivax* on their G6PD activities.

## Trial registration

The trial was registered (ACTRN12613000003774) with the Australia New Zealand Clinical trials (https://www.anzctr.org.au/Trial/Registration/TrialReview.aspx?id=363399&isReview=true).

## Author summary

At presentation of acute *Plasmodium vivax* malaria, glucose-6-phosphate dehydrogenase deficient (G6PDd) males have low G6PD activity that is unrelated to baseline reticulocyte counts; they were all detected by the qualitative fluorescent spot test. The number of G6PDd heterozygous females was too small to draw meaningful inferences. Enzyme activity rose in parallel with posttreatment reticulocytosis.

## Introduction

Glucose-6-phosphate dehydrogenase deficiency (G6PDd), the most common inherited red blood cell (RBC) enzymopathy[1], limits the rate of glutathione reduction (GSSG to 2GSH) and, consequently, the ability of RBCs to counter oxidant stress. GSH is produced by coupled redox reactions, involving the G6PD catalysed conversion of glucose-6-phosphate to 6-phospholgluco-nate in the pentose phosphate shunt, $NADP^+$ to NADPH which then regenerates GSH via glutathione reductase[2]. In G6PDd individuals, infections (e.g. pneumonia, typhoid fever), drugs [e.g. the 8-aminoquinolines, primaquine (PQ) and tafenoquine (TQ)], and fava beans [3,4,5,6,7,8] are well established causes of oxidant-related acute haemolysis (AH) that may necessitate a blood transfusion [7,9,10,11,12] and be complicated by acute kidney injury[10,13].

 PQ- and TQ-induced AH is a major concern for malaria control programmes, especially in the more severe SE Asian G6PD variants like Viangchan (a WHO Class II variant)[14] because reducing the substantial burden of vivax malaria requires attacking its hypnozoite reservoir with PQ or TQ; indeed, hypnozoites are the dominant (>80%) source of recurrent vivax infections (relapses)[15]. Accordingly, the reliable diagnosis of G6PDd is fundamental to the safe administration of the haemolytic 8-aminoquinolines.

 In this respect, a specific issue is that of women who are heterozygous for G6PD deficiency because their G6PD activity ranges from fully deficient (i.e., <30% of normal) to normal (i.e., > 80% of normal)[16]. Heterozygotes with intermediate enzyme activities (30%-70%) may screen as normal with qualitative tests and be given 8-aminoquinoline therapy with the

attendant risk of AH; two G6PD Mahidol heterozygotes were transfused following exposure to 1 mg/kg of daily PQ[11]. This risk justifies the obligatory measurement of G6PD activity before administering single-dose TQ whose long mean terminal elimination half-life of 16 days may result in prolonged AH[17].

The fluorescent spot test (FST) has been the standard qualitative screening test for the past 50 years. Although simpler and less expensive than quantitative testing, it requires a UV lamp, laboratory skills, and a cold chain for the necessary reagents, explaining partly its limited use in resource-limited, endemic settings. The FST and recently introduced, point-of-care, qualitative rapid diagnostic tests (RDTs) can reliably detect G6PD activities of < 30% activity but they have not been fully validated for detecting heterozygous females having ≥ 30% of normal activity and fail to detect G6PDd with increasing G6PD activity[18,19,20]. A 30% of normal cut off excludes effectively hemizygous males, homozygous females, and fully deficient heterozygous females from exposure to 8-aminoquinolines[21].

Where possible and practised, G6PD status is usually determined when febrile patients first present to clinics and are diagnosed with acute *P. vivax* malaria. Physicians must interpret G6PD test findings in weighing the decision to prescribe an 8-aminoquinoline. Regardless of G6PD status, acute malaria causes intra- and extravascular haemolysis and, therefore, stimulates erythropoiesis and increases the reticulocyte count. On the other hand, the reticulocyte count may be decreased through the selective destruction by *P. vivax* and disturbed erythropoiesis. This is highly relevant to G6PD status because reticulocytes have much greater (~3–8 fold) G6PD activity than mature red cells[22,23,24] and, therefore, a G6PDd patient with reticulocytosis may be misclassified as G6PDn and receive, without appropriate supervision, daily primaquine (0.25/0.5/1.0 mg/kg/d) or single-dose TQ (300 mg in an adult). Getting the G6PD diagnosis right is crucial for avoiding potential harm.

As part of a clinical trial of weekly administered primaquine to vivax-infected, G6PDd and G6PDn patients, we assessed the ability of the FST to diagnose correctly G6PDd at presentation and measured the G6PD activity and reticulocyte count at baseline and over time to ascertain how they are affected by *P. vivax* and its treatment.

## Materials and methods

### Ethics statement

Ethical approvals were obtained from the National Ethical Committee for Health Research of the Cambodian Ministry of Health, Phnom Penh (ref: 225 NECHR), and the ethical review board of the Western Pacific Regional Office of WHO, Manila, Philippine (ref: 2011. 08. CAM. 01. MVP). All patients or their legal guardians gave written or verbal consent to join this study.

The trial was registered (ACTRN12613000003774) with the Australia New Zealand Clinical trials on 3/1/13 (https://www.anzctr.org.au/Trial/Registration/TrialReview.aspx?id=363399&isReview=true).

### Trial design, study site and conduct

Study methods were detailed previously[9]. Briefly, from January 2013 to January 2014, 75 non-pregnant Cambodians aged > 1 year (y) with uncomplicated *P. vivax* were treated with dihydroartemisinin piperaquine (DHAPP, target dose of DHA 2 mg/kg/d) on Day (D) 0, 1 and 2 and eight doses (0.75 mg/kg) of weekly primaquine (D0–D49): (i) 10–17 kg, 7.5 mg, (ii) 10–25 kg, 15 mg, (iii) 26–35, 22.5 mg, (iv) 36–45 kg, 30 mg, (v) 46–55 kg, 37.5 mg, (vi) 56–75 kg, 45 mg, and (vii) ≥76 kg, 60 mg.

Key laboratory investigations were: (i) vivax parasitaemia (40 x number of vivax parasites/200 white cells on a Giemsa stained thick film), (ii) reticulocyte counts (thin blood film), (iii)

Hb concentration (HemoCue AB, Ängelholm, Sweden), (iv) Hb electrophoresis [19], (v) G6PD genotype by polymerase chain reaction [19], and (vi) G6PD activity, full blood count and routine biochemistry (D0, 7, 28, 56). G6PD status was assessed by the fluorescent spot test (FST) at baseline.

We adjusted G6PD enzyme activity in thalassaemic patients (D0, 7, 28 & 56) by taking into account their low mean corpuscular volume (MCV), which results in increased numbers of RBCs/g Hb and an artificially high G6PD activity[25] (correction factor = mean $MCV_{thalassaemia}$/ mean $MCV_{normal\ Hb}$). The lower and upper limits of normal for manually measured reticulocyte counts were defined as 0·4–2·3% [26].

## Data management and statistical methods

Clean, double entered data were analysed using Stata v14 (Stata Corporation, Texas, USA). For the enzyme analysis, four values from G6PDn patients were inappropriately low (probably related to delayed analysis) and excluded and 39 values were missing due to loss to follow up or were not done.

Proportional data between groups were compared using chi squared or Fisher's exact test, as appropriate. Normally distributed data were analysed by paired (within groups) or unpaired 't' (between groups) tests and skewed data by the corresponding non-parametric tests. Data distribution was assessed using the sktest command in Stata. The sensitivity and specificity of the FST to diagnose G6PDd patients were calculated against the G6PD genotype as the "gold standard." Using backward stepwise multivariate regression, we examined the effects of age, sex, illness days, thalassaemia, and baseline values of temperature, parasitaemia, haemoglobin and reticulocyte count on the mean baseline G6PD activity. Linear mixed effects modelling was used to assess the independent effects of age, sex, illness days, baseline splenomegaly, thalassaemia, baseline reticulocyte count and parasitaemia, the fall in Hb (baseline—nadir Hb) and G6PD activity over time. The relationship between two continuous variables was determined by simple linear regression to determine the Pearson correlation coefficient.

## Results

### Patient baseline characteristics

75 patients with microscopically-confirmed *P. vivax* mono-infections were enrolled into the study and 67 completed follow up to D56. 80% of patients were young adult males (< 30 y) and 20% were patients aged 5 –< 18y (Table 1). The 18 G6PDd patients comprised 15 hemizygous males (14 G6PD Viangchan and 1 G6PD Canton) and 3 G6PD heterozygous Viangchan females. Mean baseline Hb, reticulocyte counts, corpuscular volume and red cell distribution width were similar between the G6PDd and G6PDn arms (Table 1). With treatment, there was an initial decrease in the mean Hb, that was significantly greater (p<0.001) in the G6PDd vs. G6PDn patients (S1 Fig), followed by a sustained rise in Hb. However, one G6PDd male had a sustained fall in Hb and was transfused.

### Baseline G6PD activity and performance of the fluorescent spot test

The FST detected correctly all 15 hemizygous males and two of the three heterozygous females whilst two genotypically G6PDn patients (one female and one male) were classified as G6PD deficient for a sensitivity of 17/18, 94.4 (95% confidence interval: 72.7–99.8)% and a specificity of 55/57, 96.5 (87.9–99.6)%.

There was no correlation between the baseline G6PD activity and baseline reticulocyte counts for all patients combined, r = -0.81 (p = 0·28), and by G6PD status: (i) G6PDn r = 0.54

**Table 1. Patients' baseline characteristics.**

| Parameter | G6PD deficient n = 18 | G6PD normal n = 57 | P value |
|---|---|---|---|
| Age years | 25 (5–56) | 24 (7–63) | 0.95 |
| Aged < 18 years* | 5 (27.8) | 10 (17.5) | 0.34 |
| Male sex | 15 (83.3) | 48 (84.2) | 0.93 |
| Weight kg | 54 (20–56) | 53 (14–88) | 0.83 |
| Days ill | 2 (0–8) | 3 (0–13) | 0.14 |
| Primaquine dose mg/kg/d | 0.74 (0.65–0.78) | 0.73 (0.53–0.98) | 0.43 |
| *G6PD activity†* | | | |
| G6PD activity all patients | 0.85 (0.1–1.36) ‡ | 11.38 (6.67–16.78) | <0.0001 |
| G6PD activity % normal§ | 7.08 (0.83–11.33) | 94.83 (55.58–139.83) | |
| G6PD activity normal haemoglobin | 0.77 (0.1–1.2) | 11.23 (6.9–14.8) | |
| G6PD activity thalassaemia | 1.03 (0.72–1.36) | 11.57 (6.67–16.78) | |
| *Haemoglobin parameters* | | | |
| Haemoglobin g/dL† | 12.94 (9.6–16) | 13.26 (9–16.3) | 0.48 |
| Reticulocyte count %† | 1.86 (0.6–3.8) | 1.5 (0.5–4.5) | 0.10 |
| Normal haemoglobin | 11/17 (64.7) | 31/57 (54.4) | 0.60 |
| Heterozygous HbE | 5/17 (29.4) | 20/57 (35.1) | |
| Homozygous Hb E | 0/17 | 1/57 (1.75) | |
| Alpha thalassaemia | 1/17 (5.9) | 1/57 (1.75) | |
| Beta thalassaemia | 0/17 | 4/57 (7.1) | |
| Mean corpuscular volume (MCV) fL | 87.5 (71–97) | 84 (64–98) | 0.15 |
| High MCV (> 95 fL) | 2/16 (12.5) | 2 (3.5) | 0.20 |
| Red cell distribution width (RDW) % | 12.3 (11–16.4) | 12.9 (11–15.7) | 0.83 |
| High RDW (> 14.5%) | 2/16 (12.5) | 2/54 (3.7) | 0.22 |
| *Parasite data* | | | |
| Vivax parasitaemia/µL | 6,420 (159–9,326) | 8,300 (220–59,542) | 0.13 |

* Proportional data are shown as N (%)

† mean (range), other continuous data are median (range)

‡ includes 2 heterozygous females with measured baseline activities of 0.9 and 1.18 U / g Hb

§ In Cambodia, the median G6PD activity of a normal population is 12 U / g Hb

(p = 0.19), (ii) G6PDd r = -0.09 (p = 0.36). However, baseline G6PD activity was significantly correlated (r = 0.87, p = 0.007) with baseline temperature only in the G6PDn patients; r = 0.003 (p = 0.96) in the G6PDd patients (S2 Fig).

## Time course of G6PD activity

In both G6PDd and G6PDn patients, the mean G6PD activity followed broadly changes in the reticulocyte counts and was relatively steady, peaking on D7 [Fig 1 (males) and Fig 2 (females)]. G6PD activity increased in 42 (13 G6PDd) patients, fell in 19 (2 G6PDn) and remained unchanged in one; nine patients (21.4%) with increases in G6PD activity had concomitant decreases in reticulocyte counts (S1 Table). By linear mixed effects modelling, G6PD activity was explained only by changes in reticulocyte counts (Table 2); there was no significant effect of thalassaemia/Hb E (S3 Fig).

Compared to D0, mean D7 activities were significantly higher in the G6PDd arm (1·65 vs. 0·90 U/g Hb, p = 0·02) with a trend in the G6PDn arm (12·09 vs 11·46 U/g Hb, p = 0·06). The mean absolute changes were similar but the mean relative increases were greater in G6PDd patients: 83·3% (0·75/0·9) vs. 5·5% (0·63/11·46). One heterozygous female (Fig 2) had an

**Panel A**

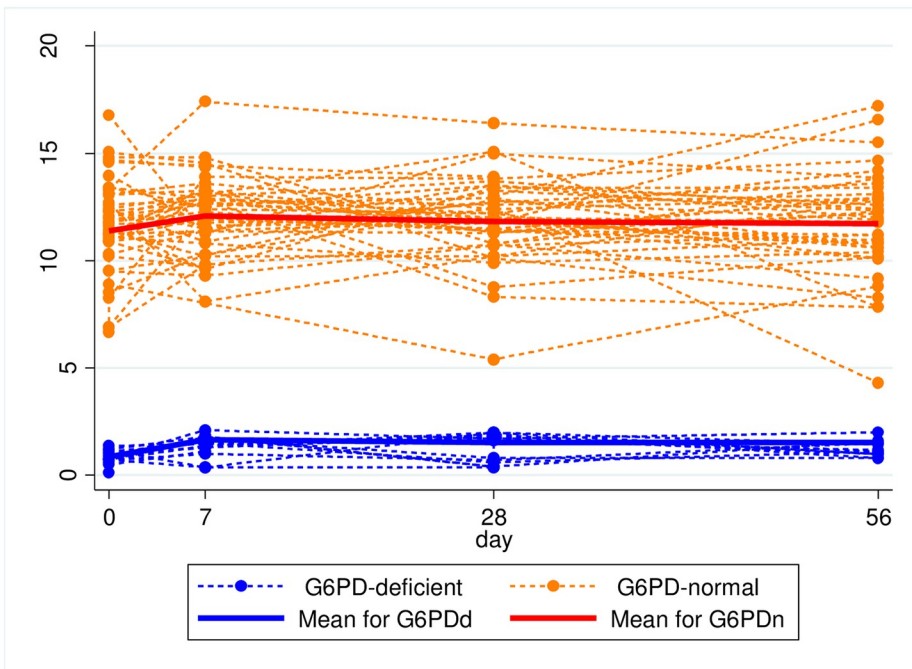

**Panel B**

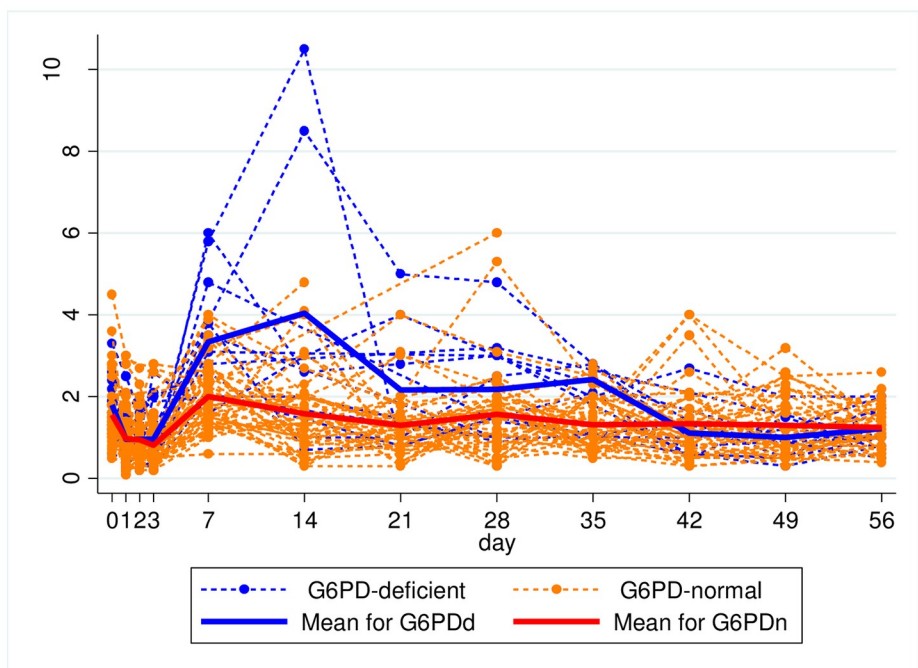

**Fig 1. Time course of G6PD activity and reticulocyte counts in male vivax-infected patients given weekly primaquine.** Panel A: G6PD activity. Panel B: reticulocyte counts.

**Panel A**

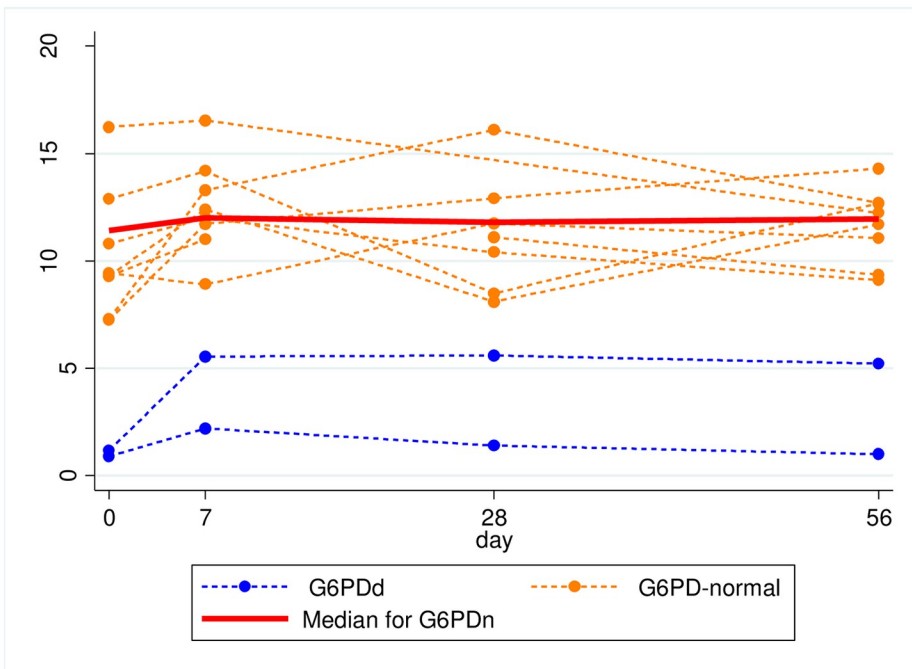

**Panel B**

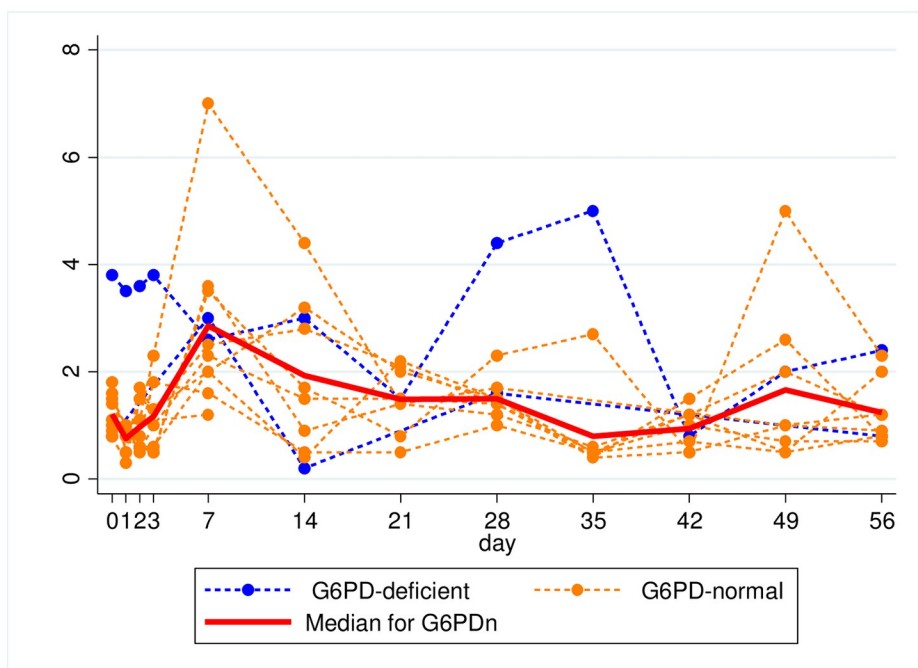

**Fig 2. Time course of G6PD activity and reticulocyte counts in female vivax-infected patients given weekly primaquine.** Panel A: G6PD activity. Panel B: reticulocyte counts.

**Table 2. Significant factors independently associated with G6PD activity over time and reticulocyte counts over time by multivariable analysis.**

| Parameter | Coefficient | 95% Confidence interval | P value |
|---|---|---|---|
| **G6PD activity over time** | | | |
| Reticulocyte count over time | 0.29 | 0.039–0.52 | 0.024 |
| **Reticulocyte counts over time** | | | |
| G6PD deficient activity over time | -0.028 | -0.051–-0.005 | 0.015 |
| Baseline reticulocyte count | 0.48 | 0.36–0.61 | <0.001 |
| Female sex | 0.34 | 0.05–0.63 | 0.021 |

absolute increase of 4·37 U/g Hb to 5·54 U/g Hb, for a 4·7-fold increase over baseline. The D7-D0 percent change in G6PD activity positively and significantly correlated (Fig 3) with the D7-D0 percent change in reticulocyte counts in the G6PDn arm (p < 0.001) but not in the G6PDd arm (p = 0.81). By D56, the mean G6PD activities were not significantly different vs. D0 (p = 0·44).

## Reticulocyte counts

The mean baseline reticulocyte count was 1·56% (range of 0·5–4·5%) and was not significantly different by G6PD status (Table 1). However, there was a significant correlation [r = 0.64 (p = 0.001)] with the baseline temperatures only in the G6PDd patients (S4 Fig). Over time, reticulocytes peaked on D14 in the G6PDd hemizygous males (Fig 1) and on D7 in the G6PDn group and in the G6PDd heterozygous females (Fig 2).

## Discussion

This is the first study detailing changes in G6PD enzyme activity in pheno- and genotypically characterised patients with *P. vivax* infection treated with single weekly PQ dose of 0·75 mg/kg for eight weeks, per a decades-old WHO recommendation. We found that baseline G6PD activity was independent of the reticulocyte count, as reported previously [27], but, over 56 days, changes in G6PD activity correlated with changes in the reticulocyte counts.

On day 7, we observed a modest increase in absolute G6PD activity in the G6PDn and G6PDd patients that correlated with an increase in reticulocyte counts although about a fifth had a concomitant fall in reticulocyte count. We interpret the reticulocytosis as resulting from two factors: reticulocyte destruction by *P. vivax* ceases and the erythropoietic activity of the bone marrow is no longer suppressed by malaria infection.

After day 7, there are fluctuations in G6PD levels in individual patients but the mean G6PD level is stable. The reticulocyte response in the G6PDd group was longer, peaking at D14, and was ~2-fold greater compared to the G6PDn group due to the additional haemolysis of vulnerable RBCs after the first and second doses of PQ.

One aim of this study was to determine whether, in a real-life situation, a G6PDd patient may be misclassified as G6PDn, because of a high baseline G6PD activity, and thus be exposed to the danger of AH when receiving a full course of PQ. Despite the limited size of the studied population, our work shows that at no point in time did the level of G6PD activity in the G6PDd hemizygous males get close to the normal range and, in contrast to the G6PDn patients, their baseline G6PD activity was unaffected by fever. Accordingly, the FST performed well in the G6PDd males with mostly G6PD Viangchan. This is consistent with several studies in SE Asia of the FST and G6PD RDTs, demonstrating their high sensitivities in malaria patients [28] and healthy individuals with enzyme activities < 30% activity [18,29] with a spectrum of mostly WHO class II G6PD variants.

**Panel A**

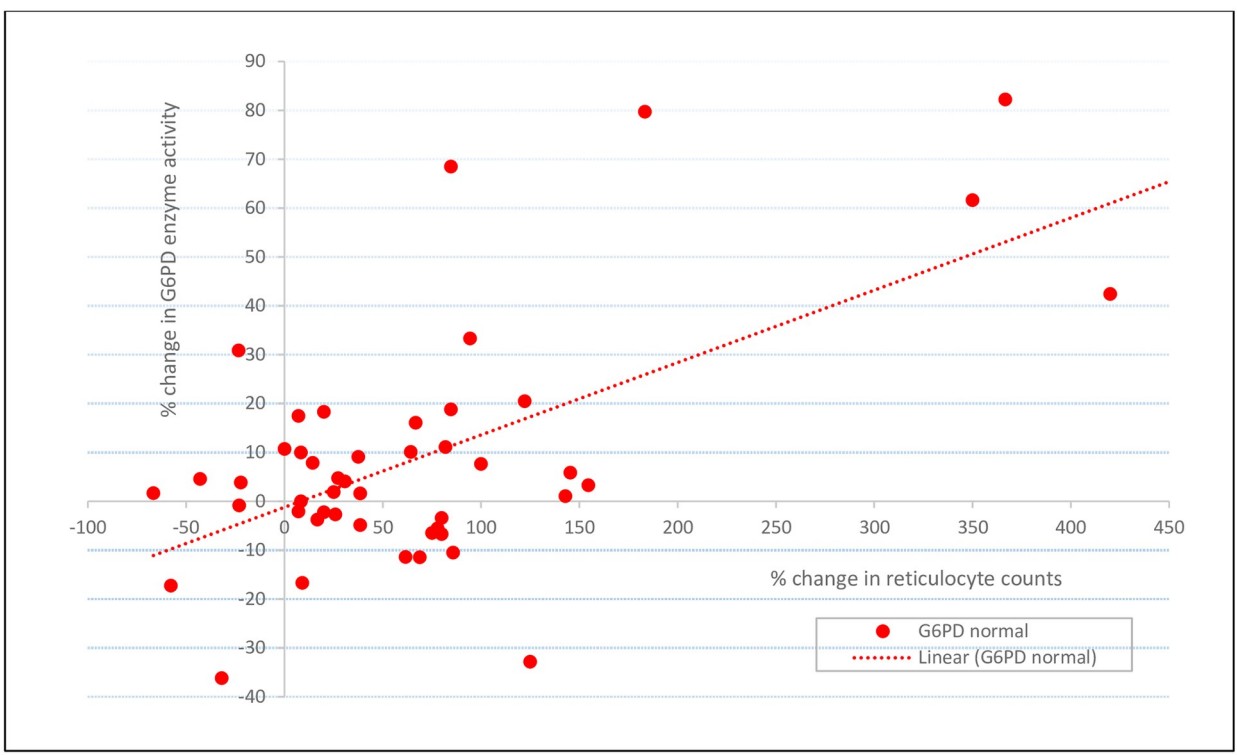

**Panel B**

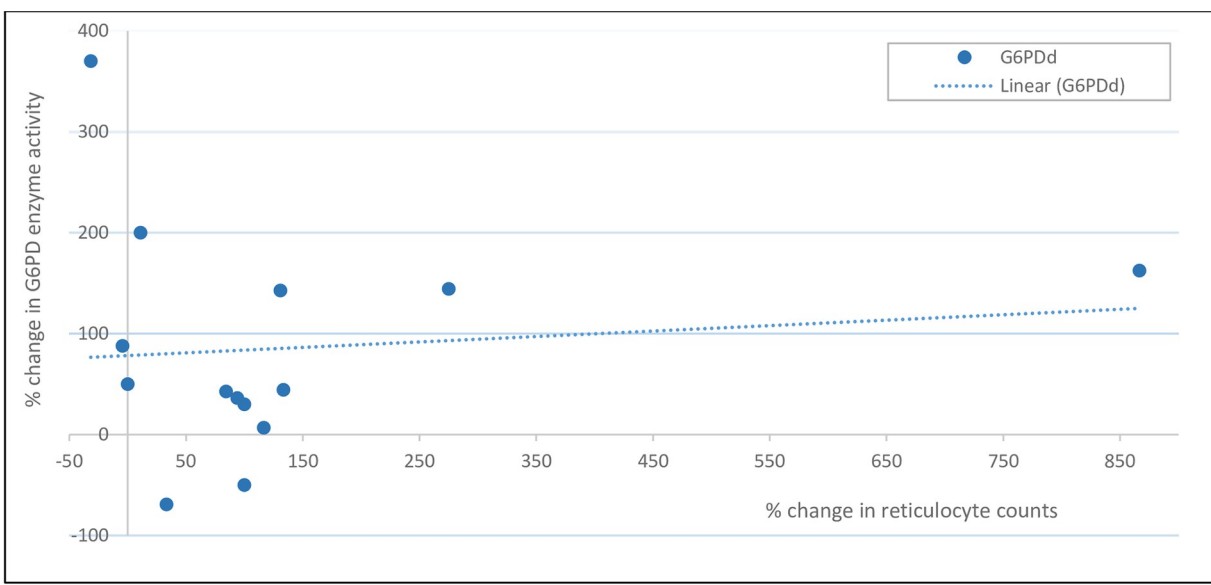

**Fig 3. Relationship between the percentage changes in G6PD activity and percentage changes in reticulocyte counts between Day 7 and baseline in the two G6PD groups.** Panel A: G6PD normal patients: the Pearson correlation coefficient, r, was 0.14, adjusted $R^2$ 0.34 (p<0.001). Panel B: G6PD deficient patients: r = 0.063, a$R^2$ = -0.063 (p = 0.64).

One of the three heterozygous females was diagnosed as normal by the FST. Her baseline G6PD activity was measured as 1·18 U/g Hb (~10% of normal) but on D7 her activity was 5·54 U/g Hb (46% of normal) and remained almost as high thereafter. Her D0 value may have been spuriously low for technical reasons and closer to her D7 value (~50% normal activity); if that were the case, the FST would be expected to classify her as normal. Other studies have reported the mis-diagnosis of G6PD heterozygous females by the FST and G6PD RDTs with declining test sensitivities, as low as ~70%, with G6PD activities between 30—< 60% in G6PDd heterozygous females [18, 28,29,30] and in an *in vitro* system of G6PD enzyme inhibition of red blood cells [20]. The skewed X-inactivation in G6PD heterozygous females results in a spectrum of phenotypes and limits the usefulness of qualitative tests and puts such women at risk of drug-induced haemolysis; only DNA analysis can identify all heterozygotes[1] but this is impractical in the field. The advent of point-of-care Biosensors that measure G6PD activity is a step forward because they can enumerate the ratio of G6PDn and G6PDd RBCs [31] in an individual heterozygote. Adequate health worker training and patient counselling on the clinical features of haemolysis are crucial when PQ, TQ and G6PD testing are rolled out.

The 8-week course of weekly PQ was associated with good Hb recovery in all but one of the G6PDd patients. We attribute this to their state of compensated haemolysis that is supported by the substantial increases (>80%) in G6PD activity, indicating a younger red cell population during this period of time, in keeping with a sustained increase in reticulocyte counts. A concomitant rise in G6PD activity and reticulocyte count has also been documented in G6PD A⁻ African Americans challenged with 30 mg of daily primaquine and that continued primaquine administration resulted in an "equilibrium/resistance" phase[32,33]. Similarly, our data suggest that the young red cells of G6PD Viangchan (and the single patient with G6PD Canton) have sufficiently high G6PD activity to confer "resistance" to haemolysis in the face of continued PQ dosing. The G6PD A- and our data contrast with the findings of Pannacciulli *et al* who showed that continued haemolysis occurred in G6PDd Mediterranean males given two courses of 30 mg of daily primaquine (30mg x7d) separated by a week to allow for a reticulocyte response[34].

Our study was limited by the small number of G6PDd patients, the very low number of G6PDd heterozygous females, almost all of whom had one G6PD variant—G6PD Viangchan. More data are needed in more G6PDd variants to support or refute our findings and provide more data on heterozygous females to guide clinicians in the field. Studies are also needed in falciparum-infected patients, given the evolving notion of giving radical cure to falciparum patients in areas of mixed vivax-falciparum transmission[35], and so determine the risk and clinical consequences of missing a diagnosis of G6PDd in patients receiving daily primaquine. We did not have a no primaquine arm which would have given us useful information on the effects of malaria itself and its treatment on the changes in G6PD activity and reticulocyte counts over time.

In conclusion, our small study of mostly G6PD Viangchang hemizygous males with acute *P. vivax* malaria have little risk of being misdiagnosed as G6PD normal by qualitative tests in our region. Although this may also apply to heterozygous females and other variants with higher mean activity e.g. G6PD A⁻, confirmation should be sought from more research.

## Supporting information

**S1 Fig. Haemoglobin changes over time in males and females.** Panel A: males. Panel B: females.
(TIF)

**S2 Fig. The relationship between the baseline G6PD activity and the baseline temperature.** A significant association was seen only in the G6PD normal patients: the Pearson correlation coefficient was 0.87 and the adjusted $R^2$ was 0.11 (p = 0.007). The Pearson correlation coefficient in the G6PD deficient patients was 0.003 (p = 0.96).
(TIF)

**S3 Fig. Glucose-6-phopshate dehydrogenase activity in patients with and without thalassaemia.** Panel A: G6PD normal patients. Panel B: G6PD deficient patients.
(TIF)

**S4 Fig. The relationship of the baseline reticulocyte count and baseline temperature in patients with acute *Plasmodium vivax* malaria by G6PD status.** The Pearson correlation coefficients were 0.64 (p = 0.001) and -0.12 (p = 0.24) in the G6PD deficient and normal patients, respectively.
(TIF)

**S1 Table. Listings of G6PD enzyme activity and their changes between Day 0 and Day 7.**
(XLSX)

## Acknowledgments

We wish to express our sincere gratitude to the patients for volunteering to join this study and to the nurses and laboratory staff.

EC was a WHO employee at the time of the study. The views expressed in the article are those of the authors and nothing to do with WHO policy.

## Author Contributions

**Conceptualization:** Walter R. J. Taylor, Saorin Kim, J. Kevin Baird, Didier Menard.

**Formal analysis:** Walter R. J. Taylor, Mavuto Mukaka.

**Funding acquisition:** Eva Christophel.

**Investigation:** Walter R. J. Taylor, Saorin Kim, Sim Kheng, Sinoun Muth, Pety Tor, Alexandra Kerleguer, J. Kevin Baird.

**Methodology:** Walter R. J. Taylor, Sim Kheng, Sinoun Muth, Pety Tor, Eva Christophel, J. Kevin Baird, Didier Menard.

**Project administration:** Saorin Kim, Sim Kheng, Sinoun Muth, Pety Tor, Didier Menard.

**Supervision:** Sim Kheng, Pety Tor, Eva Christophel, Alexandra Kerleguer, J. Kevin Baird, Didier Menard.

**Writing – original draft:** Walter R. J. Taylor, Mavuto Mukaka, Lucio Luzzatto.

**Writing – review & editing:** Saorin Kim, J. Kevin Baird, Didier Menard.

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
