## [Decision Letter · Decision Letter 0]

18 May 2021

Dear Dr. Taylor,

Thank you very much for submitting your manuscript "Dynamics of G6PD activity in patients receiving weekly primaquine for therapy of Plasmodium vivax malaria" for consideration at PLOS Neglected Tropical Diseases. As with all papers reviewed by the journal, your manuscript was reviewed by members of the editorial board and by several independent reviewers. In light of the reviews (below this email), we would like to invite the resubmission of a significantly-revised version that takes into account the reviewers' comments. 

We cannot make any decision about publication until we have seen the revised manuscript and your response to the reviewers' comments. Your revised manuscript is also likely to be sent to reviewers for further evaluation.

Sincerely,

Wuelton Marcelo Monteiro, Ph.D.

Associate Editor

Mary Lopez-Perez

Deputy Editor

Reviewer's Responses to Questions

**Key Review Criteria Required for Acceptance?**

**Methods**

-Are the objectives of the study clearly articulated with a clear testable hypothesis stated?

-Is the study design appropriate to address the stated objectives?

-Is the population clearly described and appropriate for the hypothesis being tested?

-Is the sample size sufficient to ensure adequate power to address the hypothesis being tested?

-Were correct statistical analysis used to support conclusions?

-Are there concerns about ethical or regulatory requirements being met?

Reviewer #1: The objectives of the study are not as clearly stated as desired. It is assumed (but unclear) that the primary objective is hypothesis-testing for the FST diagnosis question, but the authors do not explicitly state this. If the primary objective is hypothesis-testing, the study does not seem adequately powered and the sample population not large enough. That is, the authors claim that the FST qualitative test accurately diagnoses individuals with G6PD Viangchan but this conclusion is based on only 15 geno- and phenotypically confirmed G6PDd males and 3 heterozygous female patients. Further, the authors need to provide a justification for the sample size. The authors should consider whether or not it is even appropriate to test this hypothesis for inclusion in this manuscript. Regarding the reticulocyte portion of the study, as this is an exploratory analysis, the sample size is acceptable. Additionally, while the authors stated which statistical methods they employed, explicit mention of the assumptions tested (and how) is requested, as well as mentioning in the figure legends which statistical test was performed and for which variables the parametric assumptions were not met. Finally, critical detail on the regression model building is requested; specifically, which variables were included in the initial full model, what was the process of elimination, what was the P-value cutoff for elimination and re-entry, were there any confounding variables and if so how was it dealt with

Reviewer #2: The objectives were not clear:

1. In both the abstract and introduction, the authors state that one of the goals of the study is to “assess the impact of acute vivax malaria and of its treatment on G6PD activity levels” however, to assess the impact of vivax malaria on G6PD activity, don’t you need an uninfected comparison group? I didn’t see any data on the impact of malaria on G6PD in this study, so perhaps it should be added or this goal removed.

2. Since you are assessing the performance of a diagnostic test in the setting of an active malaria infection, would it be helpful to use ROC curves to assess the performance of the test? Sensitivity and specificity?

3. Would it be helpful to compare the performance of the test in uninfected individuals to see if it is changed by malaria infection?

4. To what extent was drug treatment versus the disease itself driving the changes in G6PD activity observed over time? Is there data available for patients who did not receive primaquine?

5. I’m surprised that G6PD activity has to be adjusted for MCV. I would expect its concentration in the cytoplasm to be the same, even if the RBC is smaller, and therefore its concentration relative to Hb would be unchanged, even if the activity per RBC is lower.

Reviewer #3: Methods are sound

**Results**

-Does the analysis presented match the analysis plan?

-Are the results clearly and completely presented?

-Are the figures (Tables, Images) of sufficient quality for clarity?

Reviewer #1: Though it is not immediately clear to this reviewer and required close scrutiny of the text and table/figure legends, the authors appear to have followed their analysis plan. However, the order of results is somewhat confusing, as the authors flip-flop the order from the Introduction. That is, in the Intro the authors first discuss the performance of the FST then the reticulocyte/G6PD activity studies but then in the Results they present the retic/G6PD activity first and FST performance second--this should be corrected. Furthermore, figures and tables require revision. First, none of the figures have titles on the y-axis; each graph should be able to stand alone. Second, this reviewer requests "A" and "B" designations for two-panel figures. Third, it is requested that all correlation coefficients regardless of statistical significance be presented in the manuscript. Fourth, it is not clear what is meant by the bolded outcome variables in Table 2--what exactly does the variable Reticulocyte Dynamics mean, is it continuous or categorical, etc. In general, it is preferred to see the marker of central tendency as median with 95% CI if the authors are concerned that their population is not normally distributed or large enough to assume normality (which it is not).

Reviewer #2: 6. Table 2 is unclear, and I could not find a definition for the factor “Reticulocyte dynamics” or the outcome variable “G6PD activity dynamics” or the outcome variable “Reticulocyte dynamics”.

Reviewer #3: Yes, except spaghetti plots are hard to look at.

**Conclusions**

-Are the conclusions supported by the data presented?

-Are the limitations of analysis clearly described?

-Do the authors discuss how these data can be helpful to advance our understanding of the topic under study?

-Is public health relevance addressed?

Reviewer #1: For the FST performance, this reviewer is concerned that the authors are over-generalizing their conclusion based on limited sample size. Furthermore, the limitations section of this study requires additional limitations and more extensive discussion. Here it may be prudent to discuss published sensitivity and specificity of the FST, whether there are known correlations between quantitative and qualitative testing (that is, at what quantitative threshold of enzyme activity does the FST correctly diagnose a patient). Discussion on this second point would be helpful for framing the FST performance results in the context of a small sample size.

Reviewer #2: 7. In the discussion, you refer to the safety and tolerability of the primaquine course and refer to one patient that needed transfusion. If safety is an endpoint in the study, please add that to the methods and results.

Reviewer #3: Yes

**Editorial and Data Presentation Modifications?**

Reviewer #1: (No Response)

Reviewer #2: 8. AHA can also refers to autoimmune hemolytic anemia, so perhaps better not to use it as an abbreviation for acute hemolytic anemia.

9. If the goal of the study was to assess the performance of the test at baseline, then are the additional time points necessary?

Reviewer #3: I'm not sure that very dense spaghetti plots are the best way to represent this data. I would favor mean or median over time for each group with error bars representing 95% CI or interquartile range. Box plots at each time point would also be much easier on the eyes.

**Summary and General Comments**

Reviewer #1: The current study addressed two clinical questions: 1) whether diagnosis by FST is impacted by acute malarial infection in G6PDd males and G6PDhet females, and 2) how do reticulocyte dynamics change and influence G6PD activity during malaria infection/treatment in G6PDd and G6PDn individuals. While the limitations of the first point are discussed below, it is worth mentioning that the retic and enzyme activity dynamics data and subsequent discussion are strong and quite compelling. This portion of the paper adds an important piece of information to the field, and is helpful for understanding the underlying retic/enzyme dynamics during acute malaria infection and treatment.

To the first question, the authors claim that the evidence presented supports the notion that diagnosis by FST is not affected. Out of 75 total patients, 15 were G6PDd males and 3 G6PDhet females, and the FST correctly diagnosed all 15 males, and of the three females, two were classified as deficient while one was diagnosed as normal, and the authors cite “technical reasons” to explain a low reading on Day 0 and that it is more likely that she was ~5 U/ g Hb. It would have been useful to include calculated sensitivities and specificities (both ~99-100%) as the major point of the study was to examine test characteristics in this population, and would have been even more helpful to include published test characteristics of FST in different G6PD variant populations, as well as the lower limit of detection of the FST test (if that data exist). That is, at what activity level (in U/ g Hb) does the FST give a qualitative diagnosis of G6PD deficiency. Even if it is true that the single patient’s Day 0 activity is falsely low, it is an over-generalization to state that this study presents clear evidence that FST does not misdiagnose G6PD heterozygous females. Perhaps the authors should consider re-framing this conclusion. Furthermore, there seems to be a disconnect between the framing of the clinical problem in the Background section and the ensuing Discussion of results. Frankly, I was expecting the sample population to be more highly enriched for G6PD heterozygous females, and I am concerned that the target population was inappropriate to answer the proposed question. While this may be due to a possible heterozygous resistance to clinically-significant malaria infection or decreased access to treatment for Cambodian women, it is a significant limitation of the study that receives inadequate discussion. If it is not possible to enrich the sample with female heterozygotes due to a low volume seen at this clinic or for whatever reason, this would be helpful to mention while responding to these comments. Further, and with regard to the presentation of results, it would have been helpful to address the major clinical question earlier in the manuscript with the reticulocytes and G6PD connection occurring later.

Regarding the reticulocyte and G6PD activity data, this was highly interesting and helpful to understand the increase in G6PD activity in the peri-infection/initiation of therapy period, though it is clear that the relationship between reticulocyte dynamics and G6PD activity is nuanced.

Specific Comments:

1. It would be helpful to designate which of the comparisons were non-parametric, and which assumptions specifically were violated. Furthermore, if the authors are significantly concerned about sample size, it might be helpful to display central tendencies as medians + 95% CIs, especially if the sample population is not normally distributed or sufficiently large enough to approach normality. In general, I would advocate for median w/ 95% CIs.

2. It would be helpful to include “A” and “B” designations for two-part figures, as well as titles for the y-axis. As it stands, no y-axis is labeled. Overall, the graphs cannot stand alone and can be difficult to interpret. 

3. Which WHO classification does the Viangchan mutation belong to? Would we expect clinical manifestations at baseline and a corresponding reticulocytosis, or were the baseline retic counts expected? It would be helpful to illuminate these expectations for the reader. 

4. The authors claim that G6PDd retics might have higher G6PD activity levels compared to G6PDn retics as evidenced by a higher relative increase in activity at Day 7 in G6PDd compared to G6PDn with a smaller increase in retic counts. Has this been reported previously or is there other evidence to support this claim?

5. Is it known what is the activity level threshold to trigger hemolysis with the study anti-malarials?

6. Typically, the abbreviation “AHA” is reserved for “autoimmune hemolytic anemia,” not “acute hemolytic anemia.” I would re-consider this usage.

7. In Line 143, is this larger drop statistically significant? While it is stated that the antimalarial therapy was well-tolerated and only one patient required transfusion, it is clinically useful to recognize the hemolytic risks of these therapies.

8. In the regression models, did the authors adjust for confounding? If so, how was this dealt with statistically? Were any indicators not included in the full regression model?

9. Table 2 is difficult to understand. It is presumed that these are three reduced (final) models and the outcome variable is in bolded text. However, it is very unclear what “reticulocyte dynamics” means in the context of the model? What type of variable is it? How was it defined? What does it mean exactly? What predictor variables were included in the initial full models?

10. In Figure 3 title, please include the correlation coefficient for G6PDd.

11. It would be of major benefit to enrich the clinical sample with female heterozygotes and would substantiate the preliminary findings from just three female hets

Reviewer #2: Summary: The authors measured the performance of a qualitative test for G6PD activity and found that it was accurate even in the setting of an active P vivax infection.

Reviewer #3: The authors have performed a study evaluating G6PD levels in subjects with acute vivax malaria and also its impact on the receipt of treatment. I have no concerns regarding the methodology, and to my knowledge this is novel. It asks and addresses a question that matters. The introduction and discussion are reasonable.

I only have one minor suggestion of relevance. I'm not sure that very dense spaghetti plots are the best way to represent this data. I would favor mean or median over time for each group with error bars representing 95% CI or interquartile range. Box plots at each time point would also be much easier on the eyes.

PLOS authors have the option to publish the peer review history of their article (what does this mean?). If published, this will include your full peer review and any attached files.

Reviewer #1: No

Reviewer #2: No

Reviewer #3: No
---

## [Editor Report · Decision Letter 1]

28 Jul 2021

Dear Dr. Taylor,

We are pleased to inform you that your manuscript 'Dynamics of G6PD activity in patients receiving weekly primaquine for therapy of Plasmodium vivax malaria' has been provisionally accepted for publication in PLOS Neglected Tropical Diseases.

Best regards,

Wuelton Marcelo Monteiro, Ph.D.

Deputy Editor

Mary Lopez-Perez

Deputy Editor

---

## [Editor Report · Acceptance letter]

3 Sep 2021

Dear Dr. Taylor,

We are delighted to inform you that your manuscript, "Dynamics of G6PD activity in patients receiving weekly primaquine for therapy of Plasmodium vivax malaria," has been formally accepted for publication in PLOS Neglected Tropical Diseases.

Best regards,

Shaden Kamhawi

co-Editor-in-Chief

Paul Brindley

co-Editor-in-Chief
